# Potomania and Beer Potomania: A Systematic Review of Published Case Reports

**DOI:** 10.3390/nu17122012

**Published:** 2025-06-16

**Authors:** Keila S. Micoanski, Jose M. Soriano, Monica M. Gozalbo

**Affiliations:** 1Unit of Psychology and Applied Nutrition, Food & Health Lab, Institute of Materials Science, University of Valencia, 46980 Paterna, Spain; keilamicoanski@outlook.com (K.S.M.); monica.gozalbo@uv.es (M.M.G.); 2Department of Medicine and Public Health, Food Science, Toxicology and Legal Medicine, University of Valencia, 46010 Burjassot, Spain; 3Joint Research Unit on Endocrinology, Nutrition and Clinical Dietetics, Health Research Institute La Fe, University of Valencia, 46026 Valencia, Spain

**Keywords:** excessive liquid intake, hyponatremia, osmotic demyelination syndrome, potomania

## Abstract

**Background/Objectives**: Potomania and beer potomania are rare but important causes of dilutional hyponatremia, resulting from excessive fluid intake combined with low solute consumption. This systematic review aimed to identify and describe the clinical presentations, underlying causes, complications, and management approaches in published case reports of these conditions. **Methods**: A systematic search was conducted in PubMed, Embase, Web of Science, and Scopus. Inclusion criteria were case reports and letters to the editor with confirmed diagnoses of potomania or beer potomania. The Joanna Briggs Institute (JBI) checklist was used to evaluate study quality. The SPIDER framework guided the selection process. A qualitative, narrative synthesis was performed. **Results**: Forty-four cases were included. Hyponatremia was the most frequent finding, commonly accompanied by neurological symptoms such as confusion and seizures. Beer potomania was more prevalent among male patients and associated with alcohol consumption and poor nutrition. Potomania was linked to restrictive diets, psychiatric disorders, or excessive intake of various non-alcoholic fluids. Management typically involved fluid restriction, correction of electrolytes, nutritional support, and psychiatric care. Five cases developed osmotic demyelination syndrome due to rapid sodium correction. **Conclusions**: Increased clinical awareness of potomania and beer potomania is essential to prevent severe outcomes. Early identification, individualized management, and cautious correction of serum sodium are crucial. Despite the limitations of case report evidence, this review provides meaningful insights into diagnosis and treatment.

## 1. Introduction

Potomania is a disorder characterized by the excessive intake of fluids without an underlying physical disorder. It can manifest as polydipsia when it refers to drinking excessive water, psychogenic polydipsia (when it is associated with mental disorders), and beer potomania when this excessive consumption is of beer [1,2]. An adult under normal physiological conditions drinks an average of 2 L of water daily, and symptoms are generally rare when fluid consumption is less than 10 L per day [3]. However, when an excess of water intake exceeds the kidneys’ elimination capacity, hypotonic hyponatremia can occur, with a reduction in plasma osmolality [4,5].

Potomania is defined as a clinical condition characterized by excessive fluid intake, particularly of liquids with low solute content, in the context of insufficient dietary protein and electrolyte consumption. This imbalance reduces renal solute availability, limiting the kidneys’ capacity to excrete free water and predisposing individuals to dilutional hyponatremia. The condition often arises from a combination of contributing factors, including maladaptive social behaviors (e.g., detox practices, competitive drinking), nutritional deficiencies, and underlying psychiatric disorders such as schizophrenia, obsessive-compulsive disorder (OCD), and bipolar disorder, which are frequently associated with psychogenic polydipsia [6,7].

Beer potomania is a specific subtype of potomania characterized by the chronic ingestion of large volumes of beer alongside a diet low in protein and electrolytes. Unlike general potomania, which may involve excessive intake of any hypo-osmolar liquid, beer potomania involves a unique combination of a solute-poor beverage and restricted solute intake from food. Since beer contains minimal electrolytes and negligible protein, it provides insufficient solute to facilitate water excretion, impairing free water clearance and leading to hyponatremia. Although both conditions share the mechanism of dilutional hyponatremia secondary to fluid overload, their etiologies, clinical contexts, and treatment approaches differ substantially [8].

The clinical importance of potomania lies in its potential to cause life-threatening hyponatremia, which can present with neurological symptoms such as seizures, altered mental status, and even osmotic demyelination syndrome if not carefully managed. Despite its severity, potomania is often underdiagnosed or misattributed to other causes of hyponatremia due to its rarity and overlapping clinical presentations. Diagnosis is further complicated by the lack of standardized criteria and reliance on patient history, which may be incomplete or unreliable, especially in those with psychiatric conditions or substance use disorders. Timely recognition and tailored interventions are critical to prevent complications and guide appropriate treatment [9,10].

In this review, cases were classified as potomania or beer potomania based on the following diagnostic criteria: (1) documented excessive intake of fluids—typically water or beer—with volumes generally exceeding 4–10 L per day; (2) presence of hypotonic hyponatremia (serum sodium < 135 mmO/L); (3) low urine osmolality (<100 mOsm/kg in most cases); (4) absence of alternative causes of hyponatremia such as renal failure, syndrome of inappropriate antidiuretic hormone secretion (SIADH), or adrenal insufficiency; and (5) dietary patterns indicating low solute or protein intake. For beer potomania, additional criteria included a reported history of chronic beer consumption with limited nutritional intake [10,11].

While individual case reports have highlighted unique presentations of potomania and beer potomania, there is a lack of synthesized evidence offering a comprehensive overview of their clinical characteristics, risk factors, and outcomes. No prior systematic review has compiled these cases to identify common patterns or to inform clinical decision-making. Given the growing recognition of fluid-induced hyponatremia and the need for awareness among clinicians, particularly in emergency and psychiatric settings, this systematic review aims to bridge the existing gap by analyzing and summarizing available evidence from case reports [12,13]. This synthesis will provide insights into diagnostic strategies, therapeutic interventions, and potential complications, supporting the development of clinical guidelines and future research directions.

## 2. Materials and Methods

To evaluate the conduct of this study and increase transparency and reproducibility, this review was registered in the PROSPERO systematic review registry database (ID: CRD42024528002). This systematic review was conducted by the Preferred Reporting Items for Systematic Reviews and Meta-Analyses (PRISMA 2020) guidelines [14]. A completed PRISMA checklist is provided as Appendix A.

Using a standardized search strategy, the databases searched for the most current data were PubMed, Embase, Web of Science, and Scopus. Boolean search strings included the following: (Potomania) OR (Beer) AND (Potomania) OR (Cider Potomania), (Potomania) AND (Anorexia), (Potomania) AND (Schizophrenia), (Potomania) AND (Exercise), (Potomania) AND (Excessive water intake), (Hyponatremia) AND (Potomania). The search included studies published up to January 2025.

Since all included studies were case reports, the SPIDER framework was used instead of PICO. The Sample included individuals with potomania or beer potomania; the Phenomenon of Interest was the clinical presentation and outcomes; the Design consisted of case reports and letters to the editor; the Evaluation involved descriptive clinical data; and the Research type was qualitative and observational.

Table 1 describes the SPIDER framework (Sample, Phenomenon of Interest, Design, Evaluation, Research type).

Inclusion criteria: (i) case reports/letters to the editor with potomania diagnosis; (ii) all age groups; (iii) clear diagnostic markers or characteristics; and (iv) reported treatment outcomes.

Exclusion criteria: (i) reviews, meta-analyses, observational or experimental studies; (ii) animal studies; (iii) irrelevant or incomplete data; (iv) languages other than English, Spanish, and French; and (v) inaccessible full texts.

Duplicates were removed with Rayyan QCR^®^. Two reviewers (K.S.M. and M.G.) independently screened records and extracted data. J.M.S resolved disagreements.

Quality was assessed with the JBI Case Report Checklist. Results are in Appendix A. No studies were excluded for low quality.

Due to descriptive data, a narrative synthesis was performed, focusing on clinical profiles, diagnostic criteria, treatments, and outcomes.

Quality assessment of the included case reports was conducted using the Joanna Briggs Institute (JBI) Critical Appraisal Checklist for Case Reports. The results of this appraisal are summarized in Appendix A and were used to guide the interpretation of findings. No cases were excluded based on low quality alone.

Because of the descriptive nature of the data, the findings were synthesized narratively. The synthesis focused on clinical characteristics, diagnostic markers, therapeutic approaches, and outcomes reported in the included case reports.

Descriptive statistics such as means, ranges, and proportions were used to summarize clinical and laboratory data reported in the case studies.

## 3. Results

A total of 44 individual cases were included in this systematic review. The mean age of patients was 45 years. Key demographic features are summarized in Table 2. Approximately 70% of the cases involved male patients. The cases were geographically diverse, with reports from Europe, North America, Asia, and South America. The detailed process of search results and selection is shown in Figure 1.

Hyponatremia was the primary clinical manifestation, with serum sodium levels ranging from 102 to 129 mmO/L. Most cases presented with severe hyponatremia (<120 mmO/L), and the most common symptoms were nausea, confusion, headache, altered mental status, seizures, and, in some cases, coma. Symptoms such as seizures and altered mental status were common, as previously described by Musch et al. [46] and Sailer et al. [47]. Fluid intake and psychiatric comorbidities are described in Table 3.

Urine osmolality was reported in 36 cases and was consistently low (around <100 mOsm/kg, referring to the average of the values found) in most patients, consistent with impaired renal capacity to excrete excess water due to low solute intake. Laboratory findings are detailed in Table 4.

Psychiatric comorbidities were present in over half of the patients, including schizophrenia, bipolar disorder, and depression. Many of these patients also had a history of psychogenic polydipsia or compulsive water intake.

Beer potomania was described in 26 of the 44 cases, predominantly affecting middle-aged males with a history of chronic alcohol consumption and poor nutritional intake. In contrast, potomania was often associated with excessive intake of other fluids (e.g., water, soda, sports drinks). It was linked to restrictive diets, eating disorders, or excessive hydration due to exercise.

Management strategies included the following:Fluid restriction (*n* = 40);Intravenous saline or hypertonic saline (*n* = 28);Electrolyte correction (e.g., potassium, magnesium, thiamine);Nutritional support;Psychiatric evaluation and intervention, where applicable.

Osmotic demyelination syndrome (ODS) was reported in five cases, all of which involved overcorrection of sodium (>10 mmO/L in 24 h). These cases emphasize the importance of cautious correction and close monitoring. Biochemical reference values used for interpretation appear in Table 5.

Outcomes were generally favorable with appropriate treatment but delayed diagnosis or overly aggressive sodium correction led to complications in a subset of patients. Mortality was reported in two cases.

The methodological quality of the included case reports was assessed using the Joanna Briggs Institute (JBI) checklist. Most cases scored well in domains such as clarity of clinical history and diagnostic reasoning. However, several reports lacked long-term follow-up or detailed outcome descriptions. These limitations were considered when interpreting the findings and drawing conclusions.

Although case reports and series are considered lower levels of evidence due to their uncontrolled nature, they play an important role in advancing medical knowledge, especially in rare disorders such as potomania. They help inform clinical decision-making when higher-level evidence is lacking. In this review, all included case reports were critically appraised using the Joanna Briggs Institute (JBI) checklist. Most studies demonstrated acceptable methodological quality, particularly in reporting clinical history and diagnostic reasoning. However, some lacked detail on treatment outcomes or long-term follow-up, and these limitations were considered in the interpretation of findings [48,49].

All included case reports listed in Table 2, Table 3 and Table 4 are cited either in the main text or referenced within the tables themselves to ensure transparency and traceability of data sources.

**Table 5 nutrients-17-02012-t005:** Reference Ranges for Relevant Biochemical Parameters.

Biochemistry References Range	
Serum sodium	133–146 mmO/L
Serum potassium	3.5–5.3 mmO/L
Serum chloride	95–108 mmO/L
Creatinine	0.6–1.2 mg/dL
Serum osmolarity	275–295 mOsmol/kg
Urine osmolality	Normal early morning urine > 600 mmol/kg
Urine sodium	40–220 mmol/24 h

References ranges were taken from Fuggle and Munro [50].

## 4. Discussion

The results of this review provide a broad overview of the clinical characteristics, manifestations, and outcomes associated with potomania and beer potomania. Across the 44 cases analyzed, distinct patterns of excessive fluid consumption, consistent clinical symptoms, and the development of hypotonic hyponatremia were observed in both conditions.

Excessive fluid intake increases total body water, dilutes plasma osmolality, and suppresses vasopressin secretion, reducing urine concentration and promoting diuresis [51]. However, when fluid intake is not accompanied by sufficient solute intake, urinary osmolality drops to around 100 mOsm/kg, limiting renal water excretion and leading to dilutional hyponatremia [52,53]. This mechanism is particularly evident in beer potomania, a condition resulting from high-volume beer consumption combined with low dietary solute and protein intake. A low urine osmolality reflects intact renal dilution capacity, but hyponatremia occurs when solute excretion is insufficient to eliminate excess water [15,54]. This dissociation between urine dilution capacity and solute-dependent water clearance has also been explored by Wilke et al. [55], who demonstrated vasopressin-independent regulation of aquaporin-2 under conditions of food deprivation.

Beer potomania was first described in the 1970s [56,57] and remains a reference for current understanding [58]. Excessive fluid intake increases total body water, dilutes plasma osmolality, and suppresses vasopressin secretion.

Additionally, very low-sodium diets prompt the kidneys to retain more sodium and water, decreasing delivery of filtrate to diluting nephron segments and further limiting water excretion—even in the absence of vasopressin [55,59]. Examples include extreme weight loss diets [23], beer potomania [54], and intake of carbohydrate-rich but sodium-poor beverages such as malted soy drinks [22], which can exacerbate hyponatremia compared to complete starvation [60].

Dietary solute intake, particularly protein and sodium, is essential for the renal excretion of free water. In patients with severely restricted diets, the low solute load impairs the kidneys’ ability to eliminate excess water, even when the capacity to dilute urine is preserved [58].

Patterns of fluid consumption and etiological context differ significantly between potomania and beer potomania. Potomania involved varied liquids, including water [61], cider [11], tea [12,13], Coca-Cola Zero^®^ [39], isotonic drinks [41], and soy-based drinks [22], while beer potomania was specifically linked to high beer consumption [62,63]. These differences reflect broader social and dietary influences, including the predominance of beer potomania in males (19 out of 23 cases), potentially due to cultural norms around alcohol consumption [60].

Symptomatically, patients with both forms of potomania presented similar clinical features: fatigue, nausea, vomiting, confusion, seizures, and altered consciousness [46,47,64]. Symptoms such as seizures and altered mental status were common, as previously described by Musch et al. [46] and Sailer et al. [47]. Potomania, schizophrenia, bipolar disorder, OCD, and substance use disorders were frequently reported [5,24,33,35], as was psychogenic polydipsia, which affects approximately 6% to 20% of patients with schizophrenia and is also prevalent in individuals with intellectual disabilities or autism spectrum disorders [65].

Diagnosis of beer potomania is based on a history of excessive beer intake, poor diet, and the exclusion of other causes of hyponatremia. It typically presents with low urine sodium and osmolality [57,58,63]. Potomania due to other fluids is diagnosed by identifying excessive intake and ruling out other etiologies of hyponatremia. Laboratory criteria for diagnosis include serum sodium < 135 mEq/L, serum osmolality < 280 mOsm/kg, and urine osmolality < 100 mOsm/kg [1].

Treatment involves fluid restriction, correction of electrolyte imbalances, and cautious use of intravenous fluids to avoid rapid sodium correction [66]. Some case reports, such as that by Yu and Fisher [16], emphasize the risks of initial fluid resuscitation in beer potomania, which may exacerbate hyponatremia if not properly monitored.

In beer potomania, alcohol cessation and nutritional support are also essential [2]. Sodium correction should not exceed 10–12 mmO/L in 24 h or 18–25 mmO/L in 48 h [54,66]. In cases of overcorrection, 5% dextrose or desmopressin may be administered [66,67]. Some case reports, such as that by Yu and Fisher [16], emphasize the risks of initial fluid resuscitation in patients with beer potomania, which may exacerbate hyponatremia if not properly monitored.

Osmotic demyelination syndrome (ODS) is a severe complication of rapid sodium correction and is characterized by symptoms such as paralysis, dysarthria, and dysphagia. Risk factors include alcoholism, malnutrition, liver disease, and correction rates > 25 mmO/L over 24–48 h [44,46,54]. Three cases in this review reported ODS [6,13,28], highlighting the need for careful management. When ODS occurs, lowering sodium using hypotonic solutions and dextrose infusions is recommended [68].

Psychiatric management also poses challenges, as several medications (e.g., lithium, carbamazepine, oxcarbazepine, valproate) may contribute to or exacerbate hyponatremia [66]. Conversely, some antipsychotics, such as risperidone and olanzapine, may reduce compulsive fluid intake behaviors [1].

Despite offering a comprehensive clinical synthesis, this study has limitations. Case reports may be subject to publication bias, with more severe or unusual presentations more likely to be reported. Additionally, the limited sample size and absence of control groups reduce the generalizability of findings. Diagnostic methods and interventions varied considerably across cases, making standardization difficult. Furthermore, sociocultural factors such as dietary habits and patterns of alcohol use may limit applicability to broader populations.

## 5. Conclusions

Potomania and beer potomania are uncommon but potentially life-threatening causes of hyponatremia, often overlooked in clinical practice. This systematic review highlights the importance of recognizing distinct fluid consumption patterns, psychiatric comorbidities, and nutritional deficiencies associated with both conditions. While potomania encompasses a broader range of liquid intakes, beer potomania is specifically linked to excessive beer consumption and low dietary solute intake.

Timely diagnosis relies on thorough patient history and laboratory evaluation, including assessment of serum sodium and serum and urine osmolality. Effective treatment includes fluid restriction, correction of electrolyte imbalances, nutritional support, and careful monitoring to prevent osmotic demyelination syndrome. Special attention should be given to psychiatric assessment and management, especially in patients with psychogenic polydipsia or other mental health conditions.

Although this review is based on case reports, which limits generalizability, it provides clinically relevant insights that can inform diagnosis and management in settings where higher-level evidence is lacking. Further research with standardized methodologies is necessary to better understand the epidemiology, outcomes, and optimal treatment approaches for these rare conditions.

## Figures and Tables

**Figure 1 nutrients-17-02012-f001:**
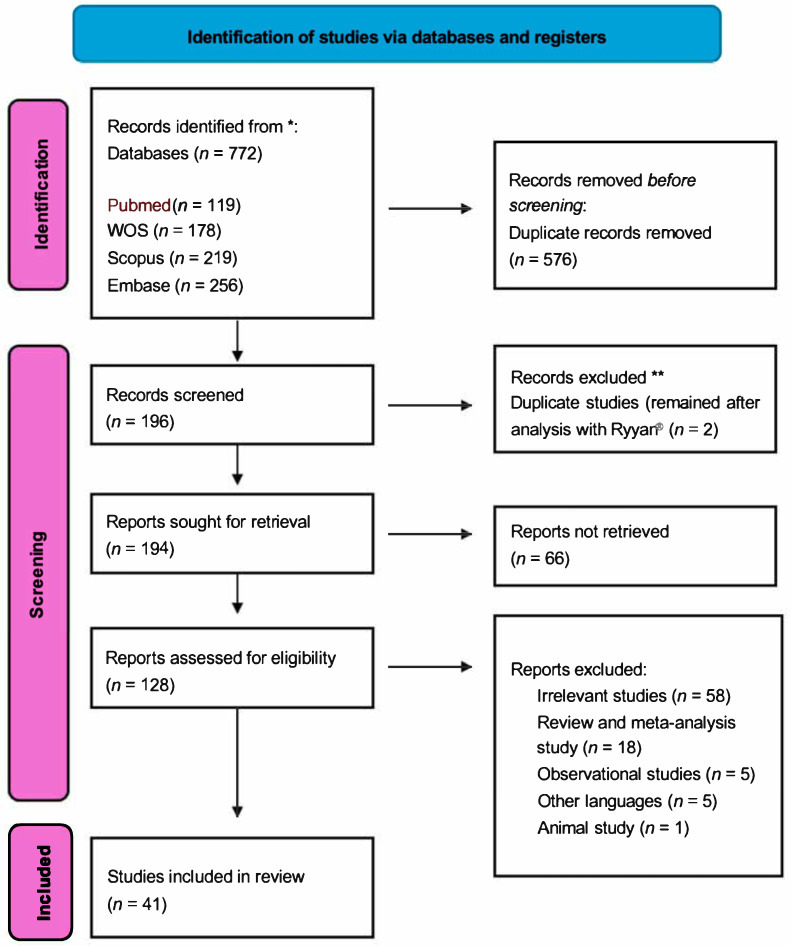
PRISMA flow diagram adapted from Page et al. [14] of the study selection process. * A total of 772 records were identified through database searching. ** After removing duplicates and applying eligibility criteria, 41 studies were included in the final review.

**Table 1 nutrients-17-02012-t001:** SPIDER Framework Used for Eligibility Criteria.

S (Sample)	Individuals diagnosed with potomania or beer potomania in published case reports.
PI (Phenomenon of Interest)	Clinical presentation, diagnostic process, treatment approaches, and outcomes related to potomania.
D (Design)	Case reports and letters to the editor describing individual or multiple clinical cases.
E (Evaluation)	Descriptive data on clinical features, laboratory findings, diagnostic tools, therapies, and patient outcomes.
R (Research Type)	Qualitative, observational evidence (non-comparative).

**Table 2 nutrients-17-02012-t002:** The description of the case reports included author and year, gender, and main diagnosis.

Author and Year	Age (Years)	Gender	Diagnosis
Lodhi et al. (2017) [15]	59 and 60	Male	Beer Potomania
Yu ZL and Fisher (2022) [16]	53	Male	Beer Potomania
Bhattarai et al. (2010) [7]	52	Male	Beer Potomania
Kujubu et al. (2015) [17]	84	Male	Beer Potomania
Senanayake et al. (2023) [10]	41	Female	Beer Potomania
Dodoo et al. (2022) [18]	56	Male	Beer Potomania
Stasishin et al. (2021) [19]	A middle-aged	Male	Beer Potomania
Laubner et al. (2020) [20]	55	Male	Beer Potomania
Mifsud et al. (2018) [21]	68	Male	Beer Potomania
So BYF and Chan GCW (2023) [22]	51	Male	Potomania
Perez Gomez et al. (2022) [8]	45	Male	Beer Potomania
Macías Robles et al. (2009) [5]	42	Male	Potomania
Soliman et al. (2018) [12]	67	Male	Potomania
Sterns et al. (2010) [23]	45	Male	Beer Potomania
Lum G. (2013) [24]	63	Male	Beer Potomania
Windpessl et al. (2017) [9]	61	Female	Potomania
Rafei et al. (2016) [25]	66	Male	Beer Potomania
Pallavi, R. (2015) [26]	68	Female	Beer Potomania
Srisung et al. (2015) [27]	69	Male	Potomania
Patel et al. (2012) [28]	56	Female	Beer Potomania
McGraw et al. (2012) [2]	Mid-30s	Male	Beer Potomania
Haran et al. (2010) [29]	41	Male	Beer Potomania
Fox BD (2002) [30]	51	Female	Potomania
Lord (2000) [11]	26	Male	Cider Potomania
Sharif et al. (2023) [31]	42	Male	Beer Potomania
Akatsuka et al. (2023) [32]	32	Male	Beer Potomania
Campeão (2013) [33]	48	Male	Potomania
Chaudhary et al. (2012) [13]	40	Female	Potomania
Axelrod et al. (2011) [34]	39	Female	Beer Potomania
Pizzini et al. (2011) [35]	76	Male	Potomania
Caron et al. (1977) [36]	Case 1: 59 Case 2: 25 Case 3: 18	Female Male Female	Potomania
Goëb et al. (2003) [37]	20	Female	Potomania
Hirel and Thobois. (2014) [38]	61	Female	Potomania
Couillard et al. (2021) [39]	54	Female	Potomania
Haymann (2014) [40]	45	Female	Potomania
Tejedor et al. (2014) [41]	41	Male	Potomania
Domínguez et al. (2013) [6]	72	Female	Potomania
Alda et al. (2007) [42]	18	Male	Potomania
Benítez-Mejía et al. (2021) [43]	52	Female	Potomania
Milisenda et al. (2012) [44]	60	Male	Beer Potomania
Campbell (2010) [45]	33	Male	Beer Potomania

**Table 3 nutrients-17-02012-t003:** Demographic and clinical characteristics of potomania participants.

Demographic and Clinical Characteristics of Potomania Participants											
Author and Year	Age	Gender	Type of Drink Consumed	Body Mass Index	Biochemical Analysis						
					Serum Sodium (mmO/L)	Serum Potassium (mmO/L)	Serum Chloride (mmO/L)	Creatinine mg/dL	Serum Osmolarity mOsm/kg	Urine Osmolality mOsm/kg	Urine Sodium (mmO/L)
So BYF and Chan GCW (2023) [22]	51 years old	Male	Malted soybean milk drink	NR	118 mmO/L	3.4 mmO/L	NR	1.76 mg/dL	NR	120 mOsm/kg	21 mmO/L
Macías Robles et al. (2009) [5]	42 years old	Male	Water	NR	98 mmO/L	1.67 mmO/L	NR	0.42 mg/dL	200 mOsm/kg	270 mOsm/kg	10 mmO/L
Soliman et al. (2018) [12]	67 years old	Male	Herbal tea (detox)	NR	111 mmol/	NR	NR	NR	NR	NR	NR
Srisung et al. (2015) [27]	69 years old	Male	Liquids	NR	117 mmOl/	NR	NR	0.7 mg/dL	250 mOsm/kg	132 mOsm /kg	NR
Campeão (2013) [33]	48-year-old male	Male	Water	NR	NR	NR	NR	NR	NR	NR	NR
Pizzini et al. (2011) [35]	76 years old	Male	Water	NR	161 mmO/L	NR	NR	4.2 mg/dL	NR	NR	NR
Caron et al. (1977) [36]	25 years old 59 years old 18 years	Male Female Female	Water	NR	139.5 mmO/L 138.5 mmO/L 142 mmO/L	4.5 mmO/L 4.5 mmO/L 4 mmO/L	NR	0.86 mg/dL 1.0 mg/dL 0.6 mg/dL	NR	NR	NR
Tejedor et al. (2014) [41]	41 years old	Male	Isotonic drinks	NR	111 mmO/L	3.1 mmO/L	79 mmO/L	NR	227 mOsm/kg	64.88 mOsm/kg	NR
Alda et al. (2007) [42]	18 years old	Male	Water	NR	NR	NR	NR	NR	NR	NR	NR
Lord (2000) [11]	26 years old	Male	Cider Potomania	NR	106 mmOl/	2.9 mmO/L	NR	0.55 mg/dL	239 mOsm/kg	NR	NR
Windpessl et al. (2017) [9]	61 years old	Female	Water and tea	19.9 kg/m^2^	122 mmO/L	3.1 mmO/L	87 mmO/L	1.1 mg/dL	251 mOsm/kg	NR	NR
Fox BD (2002) [30]	51 years old	Female	Water	33.2 kg/m^2^	108 mmO/L	2.6 mmO/L	NR	0.71 mg/dL	226 mOsm/kg	147 mOsm/kg	20 mmO/L
Chaudhary et al. (2012) [13]	40 years old	Female	Tea	NR	97 mmO/L	2 mmO/L	55 mmO/L	NR	NR	40 mOsm/kg	<10 mmO/L
Goëb et al. (2003) [37]	20 years old	Female	Water	Reported as obese	NR	NR	NR	NR	NR	NR	NR
Hirel and Thobois. (2014) [38]	61 years old	Female	Water	NR	123 mmO/L	NR	NR	NR	NR	NR	NR
Couillard et al. (2021) [39]	54 years old	Female	Coca-Cola Zero	21 kg/m^2^	147 mmol	1.26 mmO/L	NR	0.61 mg/dL	NR	NR	NR
Haymann (2014) [40]	45 years old	Female	Water	NR	103 mmO/L	4.7 mmO/L	NR	3.87 mg/dL	NR	NR	55 mmO/L
Domínguez et al. (2013) [6]	72 years old	Female	Water	NR	111 mmO/L	2.2 mmO/L	NR	NR	NR	NR	NR
Benítez-Mejía et al. (2021) [43]	52 years old	Female	Water	NR	108 mEq/L	NR	74 mmO/L	0.32 mg/dL	224 mOsm/kg	90 mOsm/kg	NR

NR: not reported.

**Table 4 nutrients-17-02012-t004:** Demographic and clinical characteristics of participants with beer potomania.

Demographic and Clinical Characteristics of Participants with Beer Potomania										
Author and Year	Age	Gender	Body Mass Index kg/m^2^	Biochemical Analysis						
				Serum Sodium mmO/L	Serum Potassium mmO/L	Serum Chloride mmO/L	Creatinine mg/dL	Serum Osmolarity mOsm/kg	Urine Osmolality mOsm/kg	Urine Sodium mOsm/kg
Lodhi et al. (2017) [15]	Case 1: 59 years old Case 2: 60 years old	Male	Case 1: 16 Case 2: 17.5 kg/m^2^	Case 1: 118 Case 2: 106 mmO/L	Case 1: 4 Case 2: 4.6 mmO/L	Case 1: 90 Case 2: 74 mmO/L	Case 1: 0.56 Case 2: 0.4 mg/dL	Case 1: 259 Case 2: 232 mOsm/kg	Case 1: - Case 2: 159 mOsm/kg	Case 1: 19 Case 2: 19 mmO/L/mEq/L
Yu ZL and Fisher (2022) [16]	53 years old	Male	Reported as thin and cachectic	113 mmO/L	4 mmO/L	70 mmO/L	0.43 mg/dL	258 mOsm/kg	94 mOsm/kg	<15 mmO/L
Bhattarai et al. (2010) [7]	52 years old	Male	NR	107 mmO/L	4 mmO/L	69 mmO/L	0.40 mg/dL	284 mOsm/kg	383 mOsm	6 mmO/L
Kujubu et al. (2015) [17]	84 years old	Male	NR	116 mmO/L	4.1 mmO/L	85 mmO/L	0.58 mg/dL	250 mOsm/kg	182 mOsm/kg	35 mEq/L
Dodoo et al. (2022) [18]	56 years old	Male	NR	102 mmO/L	4.2 mmO/L	73 mmO/L	0.3 mg/dL	245 mOsm/L	44 mOsm/L	7 mmO/L
Stasishin et al. (2021) [19]	Middle-aged	Male	NR	110 mmO/L	NR	NR	NR	NR	261 mOsm/kg	<20 mmO/L
Laubner et al. (2020) [20]	55 years old	Male	NR	105 mmO/L	4.1 mmO/L	66.9 mmO/L	0.57 mg/dL	219 mOsm/kg	NR	NR
Mifsud et al. (2018) [21]	68 years old	Male	NR	98 mmO/L	3.36 mmO/L	59 mmO/L	Reported as normal limits	218 mOsm/kg	NR	12 mmO/L
Perez Gomez et al. (2022) [8]	45 years old	Male	NR	122 ** mmO/L	NR	NR	NR	NR	NR	NR
Sterns et al. (2010) [23]	45 years old	Male	NR	96 mmO/L	3.5 mmO/L	56 mmO/L	1.0 mg/dL	203 mOsm/kg	732 mOsm/kg	7 mmO/L
Lum G. (2013) [24]	63 years old	Male	NR	106 mmO/L	4.8 mmO/L	73 mmO/L	0.8 mg/dL	245 mOsm/kg	227 mOsm/kg	<20 mmO/L
Rafei et al. (2016) [25]	66 years old	Male	NR	122 mmO/L	5 mmO/L	88 mmO/L	1 mg/dL	268 mOsm/kg	223 mOsm/kg	20 mmO/L
McGraw et al. (2012) [2]	Mid-30s	Male	16 kg/m^2^	105 mmO/L	NR	NR	NR	225 mOsm/kg	75 mOsm/kg	16 mEq/L
Haran et al. (2010) [29]	41 years old	Male	NR	94 mmO/L	3.2 mmO/L	59 mmO/L	NR	NR	NR	NR
Sharif et al. (2023) [31]	42 years old	Male	NR	97 mmO/L	NR	NR	NR	208 mOsm/kg	90 mOsm/kg	<5 mmO/L
Akatsuka et al. (2023) [32]	32 years old	Male	NR	104 mEq/L	NR	NR	17.1 mg/dL	NR	NR	NR
Milisenda et al. (2012) [44]	60 years old	Male	NR	95 mmO/L	1.7 mmO/L	NR	NR	280 mOsml	NR	NR
Campbell (2010) [45]	33 years old	Male	NR	95 mmO/L	NR	NR	NR	NR	NR	NR
Senanayake et al. (2023) [10]	41 years old	Female	18.2 kg/m^2^	121 mmO/L	5.2 mmO/L	88 mmO/L	1.02 mg/dL	265 mOsmol/kg	309 mOsmol/kg	60 mEq/day
Pallavi, R. (2015) [26]	68 years old	Female	17.5 kg/m^2^	119 mmO/L	2.8 mmO/L	NR	NR	277 mOsm/kg	79 mOsm/kg	8 mmO/L
Patel et al. (2012) [28]	56 years old	Female	NR	97 mmO/L	NR	NR	NR	NR	NR	NR
Axelrod et al. (2011) [34]	39 years old	Female	NR	NR	NR	NR	NR	NR	NR	NR

NR: not reported; ** based on estimated calculations.

## Data Availability

This review was registered in the Prospero systematic review registry database. ID: CRD42024528002.

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
