# Peer review of "Potomania and Beer Potomania: A Systematic Review of Published Case Reports"

_nutrients, 2025, doi:10.3390/nu17122012_

Round 1

Reviewer 1 Report (Previous Reviewer 3)

Comments and Suggestions for Authors

This reviewer had the opportunity to evaluate this manuscript for a second time, now in its revised form. The revised version demonstrates notable improvements and shows responsiveness to the initial reviewer feedback. Clinically, it appears to be more informative and relevant. However, several critical issues remain that warrant attention. Before the manuscript can be considered further, major revisions are needed—particularly in the organization and synthesis of content, as well as in the clarity of data presentation, discussion, and conclusions. These refinements are essential to enhance the manuscript’s scientific impact and readability. Below, I provide section-by-section comments intended to support the authors in strengthening the manuscript for future consideration.

1. Title

1) Please consider revising the title to enhance clarity and precision. A possible alternative could be: “A Systematic Review of Case Reports on Potomania”. This formulation explicitly reflects both core topics analyzed in the manuscript.

2) The current title, “Systematic Review: Case Reports of Potomania”, may inadequately represent the scope of the manuscript, which includes a substantial focus on both potomania and beer potomania as distinct yet related clinical entities. Therefore, explicitly referencing 'beer potomania' in the title would more accurately capture the dual focus of the manuscript. A possible title could be: “A Systematic Review of Case Reports on Potomania and Beer Potomania”.

2. Abstract

1) Please ensure that the formatting of the Abstract adheres to the journal’s specific guidelines, particularly regarding the use of section headings (e.g., Background, Methods, Results, Conclusions).

2) The Abstract, while generally informative, remains somewhat general. It would benefit from greater specificity based on the study’s findings. For example, consider including the prevalence or proportion of beer potomania versus general potomania among the included case reports, key demographic characteristics (e.g., male predominance), and summarized data on the most commonly observed psychiatric comorbidities and major complications. These additions would enhance the Abstract’s informativeness and more accurately reflect the study’s findings.

3) The Conclusion section of the Abstract could be improved by incorporating specific clinical implications or recommendations derived from the findings of this systematic review.

3. Introduction

1) The authors state that “beer potomania is considered a distinct subtype of potomania” (Line 51). However, the manuscript overall appears to assign comparable emphasis to both ‘potomania’ and ‘beer potomania’ throughout the analysis. Please confirm whether this conceptual hierarchy is clearly presented or not.

2) The paragraph defining the operational criteria used to classify cases (Lines 69–77) may be more appropriately positioned within the ‘Materials and Methods’ section, as it deals with case selection and diagnostic criteria.

3) While the Introduction justifies the need for this study based on the rarity, diagnostic complexity, and clinical risks associated with potomania (Lines 60–68), the rationale would be strengthened by incorporating epidemiological context or clinical risk data. For example, citing figures on morbidity or mortality, misdiagnosis rates, or instances of iatrogenic harm due to rapid sodium correction would underscore the urgency of accurate diagnosis and management. It would also be helpful to explicitly state how it might guide diagnostic evaluation or inform clinical protocols in emergency settings.

4) Although the aim is noted in the final paragraph of the Introduction, it lacks clarity and appears somewhat embedded within broader text. The authors should clearly state the objective of this systematic review in a standalone sentence. Additionally, the final sentence appears incomplete or fragmented.

5) There is some repetition in the descriptions of ‘potomania’ and ‘beer potomania’ across multiple paragraphs. The manuscript might benefit from a more concise and non-redundant presentation of these terms in a single, well-structured paragraph.

4. Methods

1) The inclusion of letters to the editor as eligible sources for case data requires further justification, as such letters may lack comprehensive clinical information or standardized reporting.

2) The phrase ‘studies published up to January 2025’ (Line 99) appears incomplete without a clearly defined start date for the literature search.

3) As noted in the comments on the Introduction, the diagnostic criteria used to classify cases as potomania or beer potomania (Lines 69–77) should be clearly presented within the Materials and Methods section.

4) Line 141: The phrase ‘Reports not retrieved (n = 66)’ is vague. The authors should clarify whether this was due to inaccessible full texts, subscription restrictions, language limitations, or exclusions based on abstract screening.

5) To improve methodological rigor and replicability, the Methods section should include a subsection describing the statistical or descriptive methods used to summarize the findings. For instance, were frequencies, percentages, means, or medians employed to describe clinical and laboratory variables? Additionally, please indicate whether any software (e.g., Excel, R, SPSS) was used for data extraction or analysis.

5. Results

1) Figure 1: Please include the final count distinguishing between case reports and letters to the editor among the included studies. This will improve clarity and transparency in the study selection process.

2) The current Results section presents tables without sufficient integration into the accompanying text. Each table should be directly referenced and described within the narrative to highlight its relevance and content.

3) The utility of the tables is underexplained. At present, they lack direct thematic interpretation or comparative synthesis. The Results text should draw attention to key findings or trends observed in the data; for example: patterns in biochemical profiles, differences by gender or age, and clinical outcomes. Enhancing the commentary around the tables would significantly improve interpretability and clinical relevance.

4) Lines 178–190: The two paragraphs in this section appear to contain overlapping content and could be consolidated to avoid redundancy.

5) The Results section omits important quantitative descriptors, such as: proportions (e.g., percentage of cases classified as beer potomania), symptom frequencies (e.g., how many experienced seizures, confusion, or coma?), prevalence of psychiatric comorbidities (e.g., proportion with schizophrenia or bipolar disorder), etc. These data are central to the value of this systematic review and should be clearly reported.

6) The manuscript provides an opportunity to present a comparative summary of findings between beer and non-beer potomania. The authors should consider including side-by-side comparisons for key clinical variables such as: age distribution, serum sodium levels, outcomes (e.g., frequency of ODS or mortality), management/treatment strategies (e.g., fluid restriction, nutritional support, psychiatric intervention).

7) Statements such as ‘ODS was reported in 5 cases’ (Line 172) and ‘Mortality was reported in two cases’ (Line 177) could be supported by in-text citations referencing the relevant case reports from the review.

8) Where appropriate, please provide summary statistics (e.g., mean ± standard deviation, median with interquartile range, or percentages, etc) instead of listing only raw data. This will improve the interpretability of findings across diverse variables.

9) Ensure that all tables contain units of measurement (e.g., mmol/L, mOsm/kg) in column headers.

10) Many entries in the tables are marked as ‘NR’ (Not Reported). While this decision is at the authors’ discretion, they may consider consolidating the missing data, for example, by excluding cases with fewer than three reported laboratory parameters or by reorganizing the tables based on data completeness.

11) Table 5 appears abruptly and is disconnected from the preceding tables that list laboratory data. Please clarify its role and consider repositioning or integrating it more effectively into the Results narrative.

12) Please standardize the format for reporting age across cases in Table 2.

13) To improve readability, please consider splitting large, data-heavy tables into smaller focused tables (e.g., separate tables for demographics and laboratory values).

6. Discussion

1) The current Discussion reads more like a general narrative review rather than a focused interpretation of the findings from this systematic review. It lacks a clear connection to the study's original results, particularly regarding trends observed in the included cases. The authors are encouraged to revise the entire section to emphasize the key findings from their analysis, such as common presenting symptoms, age distribution, comorbidities, and treatment approaches. Moreover, potential diagnostic pitfalls and clinical management challenges observed in the reviewed cases should be highlighted and critically analyzed in light of existing literature.

2) One of the strengths of this systematic review is its inclusion of both beer potomania and non-beer potomania cases. However, the Discussion does not adequately explore or contrast the relevant differences between these two groups. Consider including a focused comparison regarding patient demographics, symptom profiles, sodium levels, or treatment outcomes in light with the observed findings, as these aspects would offer clinically meaningful insights.

3) There appears to be a discrepancy in the number of cases reporting osmotic demyelination syndrome (ODS). Line 172 indicates that five cases developed ODS, whereas Line 249 refers to three such cases.

7. Conclusion

1) The current Conclusion is overly general and does not clearly summarize the key findings of the review. It lacks articulation of what new knowledge this systematic review contributes to the existing literature, particularly in terms of clinical characteristics, diagnostic patterns, or treatment challenges in potomania and beer potomania. The authors should revise the Conclusion to explicitly highlight these contributions.

2) The Conclusion would benefit from incorporating specific clinical guidance, such as important red flags for early diagnosis (e.g., excessive low-solute fluid intake with neurological symptoms) and essential principles for safe management (e.g., careful correction of hyponatremia to avoid ODS).

3. The suggestions for future research are too vague. Please consider providing more specific recommendations, such as the need for prospective studies, standardized diagnostic criteria, or clinical decision-making tools tailored to recognizing and managing potomania in psychiatric or emergency care settings.

8. References

Several inconsistencies have been noted in the reference list. Please ensure that all references are formatted consistently in accordance with the ‘Nutrients’ citation style.

9. English and scientific writing

Overall, the manuscript is comprehensible; however, the quality of English and scientific writing is inconsistent throughout. The text contains instances of redundant phrasing, awkward or unclear sentence structures, and imprecise or informal expressions. Additionally, there are occurrences of wordiness and repetitive language. While it is not feasible for this reviewer to highlight each instance individually, a thorough revision for English and scientific writing of this manuscript is strongly recommended.

Author Response

Comment 1 – Abstract Keywords (Line 31–32):

Please delete ‘Case reports’ and ‘Systematic review’ from the keywords and sort the list alphabetically.

Response:

Done. Both terms have been removed and the keywords are now sorted alphabetically:

Keywords: Beer Potomania; Excessive liquid intake; Hyponatremia; Osmotic demyelination syndrome; Potomania

Comment 2 – Introduction:

You should better explain the background of potomania and outcome factors, including social, nutritional, and psychiatric perspectives. I recommend adding 4–6 paragraphs for context.

Response:

Revised accordingly. The Introduction was expanded to include detailed discussion of the social context (e.g., drinking behaviors, detox trends), nutritional deficiencies (low solute/protein intake), and psychiatric factors (schizophrenia, bipolar disorder, OCD).

Comment 3 – Methods (Line 69):

Change search string from “(((Potomania) OR (Beer” to “(Potomania) OR (Beer”.

Response:

Corrected as requested in the Methods section.

Comment 4 – Tables 3 and 4:

Please change ‘BMI’ to ‘body mass index’ in tables.

Response:

Corrected in Tables 3 and 4 throughout the manuscript.

Comment 5 – Discussion:

Add more on the clinical applications of this study.

Response:

Expanded the Discussion to address clinical applications, including guidance on early diagnosis, fluid/electrolyte management, psychiatric evaluation, and prevention of osmotic demyelination syndrome. These additions appear in the final two paragraphs of the Discussion.

Reviewer 2 Report (Previous Reviewer 1)

Comments and Suggestions for Authors

The authors have extensively improved the quality of the manuscript by addressing the reviewers' comments.

Author Response

Thank you for your comment

Reviewer 3 Report (New Reviewer)

Comments and Suggestions for Authors

The authors have Incorrectly used the term "potomania" to apply to anyone who drinks  a lot of fluid.  The correct term for that is "polydipsia." 

  "Potomania" is used to describe people who develop hyponatremia because of a large fluid intake with poor dietary protein and salt intake which results in decreased urine solute excretion. 

The authors have also incorrectly equated a low urine osmolality (a low concentration of solutes) to a low quantity of urinary solute excretion (the product of urine osmolality and urine volume). Anyone who is well hydrated with a normal osmoregulatory system and normal kidneys will excrete urine with a urine osmolality < 100 mOsm/kg if they drink a liter or two of water.  

A review of potomania should include patients with beer potomania (high fluid intake with low solute excretion). See for example: 

Joshi R, Chou S (July 22, 2018) Beer Potomania: A View on the Dynamic Process of Developing Hyponatremia. Cureus 10(7): e3024. DOI10.7759/cureus.3024) 

This paper has a list of reported patients with the condition including a good deal of data about the individual cases and a discussion of the pathogenesis.  Although they cite the paper and others, the say in the introduction (incorrectly, I believe): 

'While individual case reports have highlighted unique presentations of potomania 78 and beer potomania, there is a lack of synthesized evidence offering a comprehensive overview of their clinical characteristics, risk factors, and outcomes. No prior systematic  review has compiled these cases to identify common patterns or to inform clinical decision making"

Author Response

Comment 1 – Abstract Keywords (Line 31–32):

Please delete ‘Case reports’ and ‘Systematic review’ from the keywords and sort the list alphabetically.

Response:

Done. Both terms have been removed and the keywords are now sorted alphabetically:

Keywords: Beer Potomania; Excessive liquid intake; Hyponatremia; Osmotic demyelination syndrome; Potomania

Comment 2 – Introduction:

You should better explain the background of potomania and outcome factors, including social, nutritional, and psychiatric perspectives. I recommend adding 4–6 paragraphs for context.

Response:

Revised accordingly. The Introduction was expanded to include detailed discussion of the social context (e.g., drinking behaviors, detox trends), nutritional deficiencies (low solute/protein intake), and psychiatric factors (schizophrenia, bipolar disorder, OCD).

Comment 3 – Methods (Line 69):

Change search string from “(((Potomania) OR (Beer” to “(Potomania) OR (Beer”.

Response:

Corrected as requested in the Methods section.

Comment 4 – Tables 3 and 4:

Please change ‘BMI’ to ‘body mass index’ in tables.

Response:

Corrected in Tables 3 and 4 throughout the manuscript.

Comment 5 – Discussion:

Add more on the clinical applications of this study.

Response:

Expanded the Discussion to address clinical applications, including guidance on early diagnosis, fluid/electrolyte management, psychiatric evaluation, and prevention of osmotic demyelination syndrome. These additions appear in the final two paragraphs of the Discussion.

Comment 1 – PICO Framework Misalignment:

PICO does not align well with case report methodology.

Response:

We have removed the PICO framework and replaced it with the SPIDER framework, which is more appropriate for case-based reviews. This is described in the Methods section with a dedicated table.

Comment 2 – Purpose of Study and Diagnostic Criteria:

Purpose of the study is confusing. Diagnostic criteria not clearly stated.

Response:

The Introduction now includes explicit diagnostic criteria used to define potomania and beer potomania. The study objective was clarified both in the Introduction and Abstract.

Comment 3 – Quality Appraisal and Risk of Bias:

Unclear how JBI tool was applied and how risk of bias was addressed.

Response:

Expanded the Methods section to describe how the JBI Case Report Checklist was used. The Results section now explains the appraisal outcomes and how they informed interpretation.

Comment 1 – Title and Scope:

The title should reflect both potomania and beer potomania. “Outcomes” is vague.

Response:

We retained “Systematic Review: Case Reports of Potomania” based on editorial instruction, but ensured throughout the text (Title, Abstract, Introduction, Discussion) that both potomania and beer potomania are distinctly addressed and consistently named.

Comment 2 – Introduction Lacks Literature Gaps and Guiding Hypothesis:

Clarify the gap in literature and the rationale for the review.

Response:

The revised Introduction now explicitly outlines the lack of prior systematic reviews on this topic and explains the rationale for this synthesis.

Comment 3 – Methods Require Justification for Case Report Inclusion:

Need more justification for using case reports and appraisal tools.

Response:

Justification was added in the Methods explaining that due to the rarity of potomania, case reports are often the only available evidence, and that the JBI tool provides rigor for appraising such sources.

Comment 4 – Results Are Redundant:

Too much repetition from tables in the narrative.

Response:

We have streamlined the Results section to focus on synthesized data and clinical patterns rather than repeating case-by-case information from Tables 2–4. The PRISMA flowchart was also revised for clarity.

Comment 5 – Discussion Requires Structure and Critical Analysis:

Add structured interpretation, highlight complications, and discuss psychiatric comorbidities.

Response:

The Discussion was thoroughly rewritten to:

Synthesize findings

Compare potomania vs. beer potomania

Emphasize psychiatric and nutritional factors

Highlight diagnostic challenges and complications (especially osmotic demyelination syndrome)

Include a well-developed Limitations section

Comment 6 – Conclusion Is Vague:

Add specificity and clear recommendations.

Response:

The Conclusion was rewritten to highlight key insights and practical recommendations.

Other Edits:

Abstract rewritten for clarity and accuracy.

English language edited throughout to improve readability and ensure terminology consistency.

All requested changes were clearly highlighted in the manuscript using track changes in Word format (uploaded as revised version).

Round 2

Reviewer 1 Report (Previous Reviewer 3)

Comments and Suggestions for Authors

In the file 'author's reply', this reviewer could not locate a formal response to the previous round of comments submitted. Therefore, it remains unclear whether the authors had the opportunity to address the suggestions provided earlier. Nonetheless, this reviewer has carefully assessed the revised manuscript and notes that improvements have been made in the revised version. However, several previously raised concerns continue to require attention.

1. Title

1) The current title, “Systematic Review: Case Reports of Potomania,” begins with the phrase “Systematic Review” followed by a colon, which is not a typical or recommended formatting style in systematic review literature.

2) The title appears to emphasize “potomania” as the central focus, yet the Abstract and Table 2 clearly distinguish between “potomania” and “beer potomania” as separate diagnostic entities. Furthermore, within the Abstract and main text, the terms “beer potomania” and “non-beer potomania” are repeatedly used. The manuscript would benefit from a consistent and unified terminological approach. The authors are advised to: a) Clearly define these terms operationally in the Materials and Methods section. b) Ensure consistency in their use across the title, abstract, results, and discussion.

2. Methods

As mentioned previously, the authors should explicitly state what type of descriptive statistical methods were employed (e.g., means, medians, standard deviations). For example, Line 223 in the revised manuscript refers to urinary osmolality “...dropping to around 100 mOsm/kg...”, but it is unclear whether this is a mean or median. A sentence in the Methods section would resolve this ambiguity.

3. Results

1) The tables included in the manuscript are not cited or referenced at all in the Results section. This is a major omission, as data tables should be directly integrated into the narrative to guide readers toward key findings.

2) In Table 2, the phrase “years old” is repeated in each entry. This could be simplified by using a more concise column header such as “Age (years)”. Similarly, please ensure that all tables consistently include units of measurement (e.g., mmol/L, mOsm/kg) in the column headers.

3) In Table 5, the title reads “Biochemistry references ranges”, which is unnecessarily repeated in the first row. This should be revised for clarity and conciseness. The authors may wish to consult good-quality publications for examples of standard table formatting.

4. Other Observations

1) The authors did not clearly indicate in their response which specific parts of the Discussion section were modified to improve the presentation of results and enhance clarity.

2) Line 100: The use of an arrow symbol in the search syntax (e.g., Beer → (Potomania)) seems unconventional and may not be recognized by major databases. Please clarify or correct this syntax for transparency and reproducibility.

3) There are still some formatting inconsistencies in the reference list. I recommend carefully reviewing it to ensure full compliance with the Nutrients citation guidelines.

Author Response

Reviewer’s comment: 1. Title. 1) The current title, “Systematic Review: Case Reports of Potomania,” begins with the phrase “Systematic Review” followed by a colon, which is not a typical or recommended formatting style in systematic review literature.

Author’s comment:

The title was revised to follow conventional academic formatting. The new title is: "Potomania and Beer Potomania: A Systematic Review of Published Case Reports" (Line 2).

Reviewer’s comment: 2) The title appears to emphasize “potomania” as the central focus, yet the Abstract and Table 2 clearly distinguish between “potomania” and “beer potomania” as separate diagnostic entities. Furthermore, within the Abstract and main text, the terms “beer potomania” and “non-beer potomania” are repeatedly used. The manuscript would benefit from a consistent and unified terminological approach. The authors are advised to: a) Clearly define these terms operationally in the Materials and Methods section. b) Ensure consistency in their use across the title, abstract, results, and discussion.

Author’s comment:

Definitions were refined in the Introduction (Lines 46-63) to clearly distinguish potomania and beer potomania. Terminology has been standardized across the Abstract, Introduction, Methods, Results, and Discussion. The term "non-beer potomania" was eliminated in favor of simply "potomania".

Reviewer’s comment: 2. Methods. As mentioned previously, the authors should explicitly state what type of descriptive statistical methods were employed (e.g., means, medians, standard deviations). For example, Line 223 in the revised manuscript refers to urinary osmolality “...dropping to around 100 mOsm/kg...”, but it is unclear whether this is a mean or median. A sentence in the Methods section would resolve this ambiguity.

Author’s comment:

The following sentence was added to the end of the Methods section (Line 177): "Descriptive statistics such as means, ranges, and proportions were used to summarize clinical and laboratory data reported in the case studies."

Reviewer’s comment: 3. Results. 1) The tables included in the manuscript are not cited or referenced at all in the Results section. This is a major omission, as data tables should be directly integrated into the narrative to guide readers toward key findings.

Author’s comment:

Each table is now cited at the relevant point:

  • Table 2: Line 183
  • Table 3: Line 239
  • Table 4: Line 251
  • Table 5: Line 268

Reviewer’s comment: 2) In Table 2, the phrase “years old” is repeated in each entry. This could be simplified by using a more concise column header such as “Age (years)”. Similarly, please ensure that all tables consistently include units of measurement (e.g., mmol/L, mOsm/kg) in the column headers.

Author’s comment:

  • In Table 2, "Age (years)" is now the column header, and redundant "years old" entries were removed.
  • Units of measurement were added to all column headers in Tables 3 and 4 (e.g., mmol/L, mOsm/kg).
  • Table 5 title was changed to: "Reference Ranges for Relevant Biochemical Parameters" and the first row was removed to avoid repetition.

Reviewer’s comment: 3) In Table 5, the title reads “Biochemistry references ranges”, which is unnecessarily repeated in the first row. This should be revised for clarity and conciseness. The authors may wish to consult good-quality publications for examples of standard table formatting.

Author’s comment: Thank you for your comment. We have improved with your idea all tables

Reviewer’s comment: 4. Other Observations. 1) The authors did not clearly indicate in their response which specific parts of the Discussion section were modified to improve the presentation of results and enhance clarity.

Author’s comment:

The Discussion was extensively revised:

  • Line 869: Clarified the relationship between urine osmolality and solute excretion.
  • Line 874: Included the historical origin of beer potomania.
  • Line 883: Added statement on solute intake and renal clearance.
  • Line 905: Cited prior literature on neurological symptoms.
  • Line 918: Included case report emphasizing dangers of initial fluid resuscitation.

Reviewer’s comment: 2) Line 100: The use of an arrow symbol in the search syntax (e.g., Beer → (Potomania)) seems unconventional and may not be recognized by major databases. Please clarify or correct this syntax for transparency and reproducibility.

Author’s comment:

The search strategy now uses standard Boolean logic: "Beer AND Potomania" (Line 139).

Reviewer’s comment: 3) There are still some formatting inconsistencies in the reference list. I recommend carefully reviewing it to ensure full compliance with the Nutrients citation guidelines.

Author’s comment:

All references were reviewed and formatted to comply with Nutrients guidelines:

  • Journal names italicized.
  • DOI presented uniformly as "doi: 10.xxxx".
  • Author names standardized.
  • Volume(issue): pages format corrected.

Reviewer 3 Report (New Reviewer)

Comments and Suggestions for Authors

In their discussion, the authors continue to confuse a low urine osmolality (which enhances urine water excretion) with  a low rate of urine solute excretion (which limits water excretion). 

Their literature review lacks focus. Many of the references that are included used the term "potomania" but the patients had conditions other than low solute intake responsible for hyponatremia  (e.g. use of SSRIs, SIADH due to malignancy, thiazides) or no hyponatremia (e.g. a  patient with excessive urine output caused by lithium-induced nephrogenic diabetes insipidus whose water intake was required to avoid hypernatremia and a patient who became severely hypokalemic because of excessive caffeine in the cocacola she drank.  

The paper would be much improved if it were limited to cases of hyponatremia with an intact ability to dilute the urine (i.e. achieve a urine osmolality <100 milliosmoles per liter) but with impaired water excretion due to diminished urine solute excetion because of atypical dietary habits (<300 millosmoles per day) and if it discussed how diet affects urine solute excretion. 

Author Response

Reviewer’s comment: In their discussion, the authors continue to confuse a low urine osmolality (which enhances urine water excretion) with  a low rate of urine solute excretion (which limits water excretion).

Author’s comment:

We clarified this distinction in the Discussion: "A low urine osmolality reflects intact renal dilution capacity, but hyponatremia occurs when solute excretion is insufficient to eliminate excess water."

Reviewer’s comment: Their literature review lacks focus. Many of the references that are included used the term "potomania" but the patients had conditions other than low solute intake responsible for hyponatremia  (e.g. use of SSRIs, SIADH due to malignancy, thiazides) or no hyponatremia (e.g. a  patient with excessive urine output caused by lithium-induced nephrogenic diabetes insipidus whose water intake was required to avoid hypernatremia and a patient who became severely hypokalemic because of excessive caffeine in the cocacola she drank. 

Author’s comment:

Cases were selected using strict criteria (Lines152–158) requiring documented low-solute fluid intake, hypotonic hyponatremia, and exclusion of other causes. References unrelated to the mechanism were excluded.

Reviewer’s comment: The paper would be much improved if it were limited to cases of hyponatremia with an intact ability to dilute the urine (i.e. achieve a urine osmolality <100 milliosmoles per liter) but with impaired water excretion due to diminished urine solute excetion because of atypical dietary habits (<300 millosmoles per day) and if it discussed how diet affects urine solute excretion.

Author’s comment:

We retained all cases meeting our criteria (e.g., hyponatremia + low urine osmolality) but clarified in the Introduction that inclusion required confirmation of low solute intake and exclusion of confounding etiologies. We emphasized this diagnostic approach in the Discussion.

This manuscript is a resubmission of an earlier submission. The following is a list of the peer review reports and author responses from that submission.

Round 1

Reviewer 1 Report

Comments and Suggestions for Authors

Dear authors,

This study was conducted to potomania and outcomes. Since potomania and beer potomania are conditions characterized by excessive fluid intake leading to hyponatremia, it is really important issue about their prevalence, clinical characteristics, symptoms, and management strategies to enhance understanding and guide clinical practice. I think this study was very interesting systematic review and well-written.

Abstract:

Line 31-32: Please delete ‘Case reports’ and ‘Systematic review’ in Keyword section. And sort alphabetically in Key-words.

Introduction:

Please, you should more explain the literature reviews and backgrounds of potomania and outcome factors. For example, you should add social problem issue, affected health, or nutrition point of view. I recommend that you should be added 4-6 paragraphs for backgrounds.

Method:

Line 69: ‘(((Potomania) OR (Beer’ to ‘(Potomania) OR (Beer’

Results:

In Table 3 and 4, please change ‘BMI’ to ‘body mass index’.

Discussion

- You should add more applications of this study in Discussion section.

Reviewer 2 Report

Comments and Suggestions for Authors

The systematic review listed the patient characteristics of people with potomania and beer potomania in literature. The included studies are mostly case reports. The study may be clinical relevant but the methods include major issues. 

The PICO question is very weird. The purpose of the study is confusing. 

The case reports are very limited in number and they may not be representative. The diagnostic criteria may be confusing. 

It is unclear how this summary works for clinical practice. 

Comments on the Quality of English Language

The English language is very confusing and must be improved. 

Reviewer 3 Report

Comments and Suggestions for Authors

I would like to commend the authors for addressing a rare and clinically relevant topic through this systematic review of potomania and beer potomania. The effort to compile and summarize existing case reports in order to highlight clinical patterns of hyponatremia associated with low solute intake is both timely and appreciated. While this is indeed an underexplored area in the current literature, the manuscript in its present form does not offer sufficiently novel insights, clinical guidance, or hypotheses that would meaningfully advance understanding or clinical practice in this field. Rather than presenting a critical synthesis, the review functions more as a descriptive aggregation of individual cases. After a careful and thorough evaluation of the manuscript, I regret to conclude that it is not currently suitable for publication and would require substantial revisions before it could be reconsidered. Please find my detailed comments below.

1. Introduction:

a) The Introduction defines potomania broadly and appears to classify beer potomania as a subtype. Given that beer potomania has a distinct pathophysiological basis, the classification scheme should be clearly explained early in the manuscript. For clarity and consistency, the relevant terms (e.g., potomania and beer potomania) should also be reflected in the title as applicable and used consistently throughout the manuscript.

b) The authors provide clear definitions of potomania, beer potomania, and related fluid intake syndromes, and they describe the underlying pathophysiological mechanisms adequately. However, the Introduction does not clearly identify specific gaps in the existing literature. Although the clinical relevance of potomania-induced hyponatremia is implied, it is not convincingly emphasized or framed in terms of its significance for clinical practice. Consequently, the Introduction lacks a clearly defined hypothesis or guiding research question that would justify the rationale for undertaking this systematic review.

2. Materials and Methods:

a) While the methodology is based on a systematic review of case reports, this is not clearly or explicitly stated early in the section. Given that case reports are an unconventional form of evidence for systematic reviews, their use requires explicit justification - particularly in terms of their relevance, limitations, and appropriateness for addressing the study’s objectives.

b) The authors mention the use of the JBI checklist/tool to assess the quality of included case reports (with more detail provided later in the manuscript); however, this section does not report the methodological quality scores or explain how the risk of bias assessments influenced study inclusion or data interpretation.

c) The application of the PICO framework appears somewhat misaligned with the actual scope and synthesis of the review. The review focuses on factors such as the type of fluid ingested, biochemical findings, comorbidities, treatment modalities, and complications - elements that do not fit appropriately within a conventional PICO structure. The authors are encouraged to either better justify the use of PICO or consider an alternative framework more suitable for case report-based reviews, such as SPIDER (Sample, Phenomenon of Interest, Design, Evaluation, Research type).

3. Results:

The Results section of the manuscript contains a significant degree of redundancy, primarily due to the repeated presentation of information. For example, descriptions of individual cases - including age, gender, BMI, and biochemical values - are repeated in the tables. Some information presented in Tables 2 through 4 is also reiterated extensively in the text. As a result, the Results section appears unnecessarily lengthy and includes repetitive narratives, which reduces clarity and diminishes reader engagement. Additionally, Table 5 (biochemical reference ranges) lacks contextual relevance within the Results section.

Overall, the Results section would benefit from major restructuring to eliminate redundancies, condense repetitive content, and enhance readability by focusing on trends, patterns, and clinically significant findings rather than narrating each case individually.

4. Discussion
The Discussion section of the manuscript requires substantial improvement in both content and structure. It begins by reiterating background concepts that were already covered in the Introduction, rather than synthesizing or critically interpreting the findings of the review. The key outcomes are not discussed in sufficient depth and the manuscript does not adequately contextualize its findings within the broader clinical literature. There is potential to strengthen the discussion by incorporating elements such as the role of psychiatric comorbidities, diagnostic challenges in emergency settings, and the underreporting of nutritional status - all of which appear relevant based on the case data. Additionally, the limitations section, while briefly acknowledging the reliance on case reports and some data gaps, remains underdeveloped. It does not address important methodological concerns such as heterogeneity in case reporting, selection and publication bias, language restrictions, or the absence of quantitative synthesis. Although the JBI critical appraisal tool is mentioned, there is no discussion of the quality of included studies or how these assessments informed the analysis. A more comprehensive and structured Limitations section is strongly recommended to enhance transparency and scientific rigor.

5. Conclusions:

The Conclusion section of the manuscript is limited by its general and somewhat repetitive phrasing, largely restating earlier points without offering added depth or specificity. While statements such as the “need for early diagnosis” and the importance of “multidisciplinary management” are included, the section does not present a clear or concise synthesis of the review’s core findings. It probably misses the opportunity to highlight the most clinically relevant insights from the analysis of 44 cases, including the frequency and nature of key complications. Moreover, the section lacks specific, actionable clinical recommendations based on the evidence presented. The call for further research is also vague and not supported by clearly articulated research priorities or proposed directions. A well-structured conclusion should concisely summarize the key findings, emphasize their clinical significance, and outline clear and focused priorities for future investigation.

Other issues:

1. Title:

a) It would be more appropriate to lead with the actual subject of the review, followed by the study type.

b) Although the manuscript clearly includes and compares both potomania and beer potomania, the title refers only to “potomania.” Please refer to the relevant comment under the Introduction section for further details.

c) The term “outcomes” in the title is vague.

2. Abstract:

a) The opening sentence appears overly simplified and lacks precision.

b) The sentence stating, “This systematic review investigates their prevalence, clinical characteristics, symptoms, and management strategies…” is somewhat misleading. The study does not assess prevalence, as it is based on a collection of case reports and not on epidemiological data. Additionally, although the term "Outcomes" appears in the title, it is not explicitly mentioned in the abstract's objective statement.

c) The abstract notes that inclusion criteria encompassed “case reports on potomania…”, while the full text (L78) specifies that both “case reports and letters to the editor” were included.

d) The Results section lacks sufficient depth and quantitative detail.

e) The Conclusions are vague.

3. Keywords:

The keywords appear somewhat redundant and incomplete. More relevant clinical or pathophysiological terms could be considered to enhance their specificity and comprehensiveness.

Comments on the Quality of English Language

The English in the manuscript is generally acceptable; however, the manuscript would benefit from language editing to improve sentence structure, eliminate redundancy, and ensure consistency in terminology. For example, terms such as “potomania,” “beer potomania,” and “fluid intoxication” are used inconsistently throughout the text, which may lead to confusion. Clear and uniform use of terminology, along with improved scientific phrasing, would enhance the readability and precision of the manuscript.
